

# Aggregating pairwise semantic differences for few-shot claim verification

Xia Zeng and Arkaitz Zubiaga

Queen Mary University of London, London, United Kingdom

## ABSTRACT

As part of an automated fact-checking pipeline, the claim verification task consists in determining if a claim is supported by an associated piece of evidence. The complexity of gathering labelled claim-evidence pairs leads to a scarcity of datasets, particularly when dealing with new domains. In this article, we introduce Semantic Embedding Element-wise Difference (SEED), a novel vector-based method to few-shot claim verification that aggregates pairwise semantic differences for claim-evidence pairs. We build on the hypothesis that we can simulate class representative vectors that capture average semantic differences for claim-evidence pairs in a class, which can then be used for classification of new instances. We compare the performance of our method with competitive baselines including fine-tuned Bidirectional Encoder Representations from Transformers (BERT)/Robustly Optimized BERT Pre-training Approach (RoBERTa) models, as well as the state-of-the-art few-shot claim verification method that leverages language model perplexity. Experiments conducted on the Fact Extraction and VERification (FEVER) and SCIFACT datasets show consistent improvements over competitive baselines in few-shot settings. Our code is available.

# INTRODUCTION

As a means to mitigate the impact of online misinformation, research in automated fact-checking is attracting increasing attention (*Zeng, Abumansour & Zubiaga, 2021*). A typical automated fact-checking pipeline consists of two main components: (1) claim detection, which consists of identifying the set of sentences out of a long text deemed capable of being fact-checked (*Konstantinovskiy et al., 2020*), and (2) claim validation, which aims to do both evidence retrieval and claim verification for claims (*Pradeep et al., 2020*).

As a key component of the automated fact-checking pipeline, the claim verification (the task is sometimes referred to as veracity classification (*Lee et al., 2021*)) component is generally framed as a task in which a model needs to determine if a claim is supported by a given piece of evidence (*Thorne et al., 2018*; *Wadden et al., 2020*; *Lee et al., 2021*). It is predominantly tackled as a label prediction task: given a claim $c$ and a piece of evidence $e$, predict the veracity label for the claim $c$ which can be one of "*Support*", "*Contradict*" or "*Neutral*". The Fact Extraction and VERification (FEVER) (*Thorne et al.,*

Corresponding author
Xia Zeng, x.zeng@qmul.ac.uk

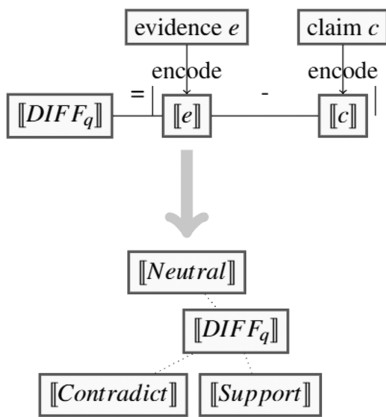

**Figure 1** SEED consists of two steps: 1. Captures average semantic differences between claim-evidence pairs for each class, leading to a $[\![DIFF_q]\!]$ representative vector per class. 2. During inference, each input vector $[\![DIFF_q]\!]$ is compared with these representative vectors.

*2018*) dataset presents the following example: the claim, "A staging area is only an unused piece of land." is contradicted by the evidence, "A staging area (otherwise staging point, staging base or staging post) is a location where organisms, people, vehicles, equipment or material are assembled before use".

Despite recent advances in the claim verification task, existing methods predominantly involve training big language models, and/or rely on substantial amounts of labelled data, which can be unrealistic in the case of newly emerging domains such as COVID-19 (*Saakyan, Chakrabarty & Muresan, 2021*). To overcome these dependencies, we set out to propose a novel and effective method to claim verification with very limited data, *e.g.*, as few as 10 to 20 samples per veracity class. To develop this method, we hypothesise that a method can leverage a small number of training instances, such that the semantic differences will be similar within each veracity class. Hence, we can calculate a representative vector for each class by averaging semantic differences within claim-evidence pairs of that class. These representative vectors would then enable making predictions on unseen claim-evidence pairs. Figure 1 provides an illustration.

Building on this hypothesis, we propose a novel method, Semantic Embedding Element-wise Difference (SEED), as a method that can leverage a pre-trained language model to build class representative vectors out of claim-evidence semantic differences, which are then used for inference. By evaluating on two benchmark datasets, FEVER and SCIFACT, and comparing both with fine-tuned language models, Bidirectional Encoder Representations from Transformers (BERT) (*Devlin et al., 2019*) and Robustly Optimized BERT Pre-training Approach (RoBERTa) (*Liu et al., 2019*), and with the state-of-the-art few-shot claim verification method that leverages perplexity (*Lee et al., 2021*), we demonstrate the effectiveness of our method. SEED validates the effectiveness of our proposed paradigm to tackle the claim verification task based on semantic differences, which we consistently demonstrate in three different settings on two datasets.

In this article, we conduct comprehensive experiments in order to answer two overarching research questions:

- **RQ1:** Can sentence embeddings from pre-trained language models be effectively utilised to compute pairwise semantic differences between claims and their associated evidences with limited labelled instances?
- **RQ2:** If so, would they contribute positively to the task of claim verification in few-shot settings?

We make the following contributions:

- We introduce SEED, a novel method that computes semantic differences within claim-evidence pairs for effective and efficient few-shot claim verification.
- By experimenting on two datasets, we demonstrate the effectiveness of SEED to outperform two competitive baselines in the most challenging settings with limited numbers of shots. While the state-of-the-art perplexity-based model is restricted to two-class classification, SEED offers the flexibility to be used in two- or three-class settings. By looking at classwise performance results, we further demonstrate the consistent improvement of SEED across all classes.
- We perform a *post-hoc* analysis of the method, further delving into the results to understand performance variability through standard deviations, as well as to understand method convergence through the evolution of representative vectors.

## BACKGROUND

When dealing with claim verification, most recent systems fine-tune a large pre-trained language model to do three-way label prediction, including VERISCI (*Wadden et al., 2020*), VERT5ERINI (*Pradeep et al., 2020*), and ParagraphJoint (*Li, Burns & Peng, 2021*). Despite the evident effectiveness of these methods, fine-tuning models depends on the availability of substantial amounts of labelled data, which are not always accessible, particularly for new domains. They may also be very demanding in terms of computing resources and time. Given these limitations, here we argue for the need of developing more affordable solutions which can in turn achieve competitive performance in few-shot settings and/or with limited computing resources.

Research in few-shot claim verification is however still in its infancy. To the best of our knowledge, existing work has limited its applicability to binary claim verification, *i.e.*, keeping the *"Support"* class and merging the *"Contradict"* and *"Neutral"* classes into a new *"Not_Support"* class. *Lee et al. (2021)* hypothesised that evidence-conditioned perplexity score from language models would be helpful for assessing claim veracity. They explored using perplexity scores with a threshold *th* to determine claim veracity into *"Support"* and *"Not_Support"*: if the score is lower than the threshold *th*, it is classified as *"Not_Support"* and otherwise *"Support"*. This method proved to achieve better performance on few-shot binary classification than fine-tuning a BERT model. In proposing our SEED method, we use this method as the state-of-the-art baseline for

few-shot claim verification in the same two-class settings, while SEED is also applicable to and experimented in three-class settings.

Use of class representative vectors for text classification has also attracted interest in the research community recently. In a similar vein to our proposed approach SEED, prototypical networks (*Snell, Swersky & Zemel, 2017*) have proven successful in few-shot classification as a method using representative vectors for each class in classification tasks. Prototypical networks were proposed as a solution to iteratively build class prototype vectors for image classification through parameter updates *via* stochastic gradient descent, and have recently been used for relation extraction in NLP (*Fu & Grishman, 2021*, *Gao et al., 2019*). While building on a similar idea, our SEED method further proposes the use of semantic differences to simulate a meaningful and comparable representation of claim-evidence pairs, enabling its application on the task of claim verification.

## SEED: METHODOLOGY

We hypothesise that we can make use of sentence embeddings from pre-trained language models such as BERT and RoBERTa to effectively compute pairwise semantic differences between claims and their associated evidences. These differences can then be averaged into a representative vector for each class, which can in turn serve to make predictions on unseen instances during inference.

We formalise this hypothesis through the implementation of SEED as follows. For a given pair made of *claim* and *evidence*, we first leverage a pre-trained language model through sentence-transformers library (*Reimers & Gurevych, 2019*) to obtain sentence embeddings $[\![claim]\!]$ and $[\![evidence]\!]$. Specifically, embeddings are obtained by conducting mean pooling with attention mask over the last hidden state. We then capture a representation of their semantic difference by calculating the element-wise difference $|[\![claim]\!] - [\![evidence]\!]|$. To the best of our knowledge, its previous implementation is only found in *Reimers & Gurevych (2019)* as one of many available classification objective functions, leaving room for further exploration. Formally, for a claim-evidence pair $i$ that has $evidence_i$ and $claim_i$, we have Eq. (1):

$$[\![DIFF_i]\!] = |[\![evidence_i]\!] - [\![claim_i]\!]| \tag{1}$$

To address the task of claim verification that compares a claim with its corresponding evidence, we obtain the mean vector of all $[\![DIFF]\!]$ vectors within a class. We store this mean vector as the representative of the target claim-evidence relation. That is, for each class $c$ that has $n$ training samples available, we obtain its representative relation vector with Eq. (2).

$$\begin{aligned}
[\![Relation_c]\!] &= [\![\overline{DIFF_c}]\!] \\
&= \frac{1}{n}\sum_{i=1}^{n}([\![DIFF_i]\!]) \\
&= \frac{1}{n}\sum_{i=1}^{n}(|[\![evidence_i]\!] - [\![claim_i]\!]|)
\end{aligned} \tag{2}$$
During inference, we first obtain the query $\llbracket DIFF_q \rrbracket$ vector for a given unseen claim-evidence pair, then calculate Euclidean distance between the $\llbracket DIFF_q \rrbracket$ vector and every computed $\llbracket Relation_c \rrbracket$ vector, e.g., $\llbracket Support \rrbracket$, $\llbracket Contradict \rrbracket$ and $\llbracket Neutral \rrbracket$ for three-way claim verification, and finally inherit the veracity label from the candidate relation vector that has the smallest Euclidean distance value.

# EXPERIMENT SETTINGS

## Datasets

We conduct experiments on the FEVER (*Thorne et al., 2018*) and SCIFACT (*Wadden et al., 2020*) datasets (see examples in Table 1). FEVER, a benchmark, large-scale dataset for automated fact-checking, contains claims that are manually modified from Wikipedia sentences and their corresponding Wikipedia evidences. SCIFACT is a smaller dataset that focuses on scientific claims. The claims are annotated by experts and evidences are retrieved from research article abstracts. For notation consistency, we use *"Support"*, *"Contradict"* and *"Neutral"* as veracity labels for both datasets[1].

## Method implementation

We implement SEED using the sentence-transformers library (*Reimers & Gurevych, 2019*) and the huggingface model hub (*Wolf et al., 2020*). Specifically, we use three variants of BERT (*Devlin et al., 2019*) as the base model: BERT-base, BERT-large and BERT-nli[2]. We include experiments with $SEED_{BERT_{NLI}}$ due to the proximity between the claim verification and natural language inference tasks. We use $SEED_{BERT_B}$, $SEED_{BERT_L}$ and $SEED_{BERT_{NLI}}$ to denote them hereafter.

## Baselines

We compare our method with two baseline methods: perplexity-based (PB) method and fine-tuning (FT) method.

### Perplexity-based method (PB)

The perplexity-based method (*Lee et al., 2021*) is the current SOTA method for few-shot claim verification. It uses conditional perplexity scores generated by pre-trained language models to find a threshold that enables binary predictions. If the perplexity score of a given claim-evidence pair is higher than the threshold, it is assigned the "Support" label; otherwise, the "Not_Support" label. We conduct experiments with BERT-base and BERT-large for direct comparison with other methods. We denote them as $PB_{BERT_B}$ and $PB_{BERT_L}$ hereafter.

### Fine-tuning method (FT)

We also conduct experiments with widely-used model fine-tuning methods. Specifically, we fine-tune vanilla BERT-base, BERT-large, RoBERTa-base and RoBERTa-large models[3]. Following (*Lee et al., 2021*), we use $5e^{-6}$ for $FT_{BERT_B}$ and $FT_{RoBERTa_B}$ as learning rate and $2e^{-5}$ for $FT_{BERT_L}$ and $FT_{RoBERTa_L}$. All models share the same batch size of 32 and are trained for 10 epochs. We denote them as $FT_{BERT_B}$, $FT_{BERT_L}$, $FT_{RoBERTa_B}$ and $FT_{RoBERTa_L}$ hereafter.

[1] Originally, FEVER uses *"Support"*, *"Refute"* and *"Not Enough Info"* as veracity categories, while SCIFACT uses *"Supports"*, *"Refutes"* and *"No Info"*.

[2] The first two are available on huggingface model hub (*Wolf et al., 2020*) with model id *bert-base-uncased* and bert-largeuncased. The last one has been fine-tuned on natural language inference (NLI) tasks and is available on sentence-transformers repository with model id *bert-base-nli-mean-tokens*.

[3] The associated model ids from huggingface model hub (*Wolf et al., 2020*) are *bert-base-uncased, bert-large-uncased, robertabase* and *roberta-large* respectively.

**Table 1 Veracity classification samples from the FEVER (*Thorne et al., 2018*) and SCIFACT (*Wadden et al., 2020*) datasets.**

**FEVER**

| Claim | Evidence | Veracity |
|---|---|---|
| "In 2015, among Americans, more than 50% of adults had consumed alcoholic drink at some point." | "For instance, in 2015, among Americans, 89% of adults had consumed alcohol at some point, 70% had drunk it in the last year, and 56% in the last month." | "Support" |
| "Dissociative identity disorder is known only in the United States of America." | "DID is diagnosed more frequently in North America than in the rest of the world, and is diagnosed three to nine times more often in females than in males." | "Contradict" |
| "Freckles induce neuromodulation." | "Margarita Sharapova (born 15 April 1962) is a Russian novelist and short story writer whose tales often draw on her former experience as an animal trainer in a circus." | "Neutral" |

**SCIFACT**

| Claim | Evidence | Veracity |
|---|---|---|
| "Macropinocytosis contributes to a cell's supply of amino acids *via* the intracellular uptake of protein." | "Here, we demonstrate that protein macropinocytosis can also serve as an essential amino acid source." | "Support" |
| "Gene expression does not vary appreciably across genetically identical cells." | "Genetically identical cells sharing an environment can display markedly different phenotypes." | "Contradict" |
| "Fz/PCP-dependent Pk localizes to the anterior membrane of notochord cells during zebrafish neuralation." | "These results reveal a function for PCP signalling in coupling cell division and morphogenesis at neurulation and indicate a previously unrecognized mechanism that might underlie NTDs." | "Neutral" |

## Experimental design

Experiments are conducted in three different configurations: binary FEVER claim verification, three-way FEVER claim verification and three-way SCIFACT claim verification. The first configuration is designed to enable direct comparison with the SOTA method (*i.e.*, PB), as it is only designed for doing binary classification.

We conduct n-shot experiments ($n$ training samples per class) with the following choices of $n$: 2, 4, 6, 8, 10, 20, 30, 40, 50, 100. Note that one may argue that 50-shot and 100-shot are not necessarily few-shot, however we chose to include them to further visualise the trends of methods up to 100 shots. The number of shots $n$ refers to the number of instances per class, *e.g.*, 2-shot experiments would include six instances in total when experimenting with three classes. To control for the performance fluctuations owing to the randomness of shots selection, we report the mean results for each $n$-shot experiment obtained by using 10 different random seeds ranging from 123 to 132. Likewise, due to the variability in performance of the FT method given its non-deterministic nature, we do five runs for each setting and report the mean results.

## RESULTS

We first report overall accuracy performance of each task formulation, then report classwise F1 scores for three-way task formulations. Finally we report statistical significance results.

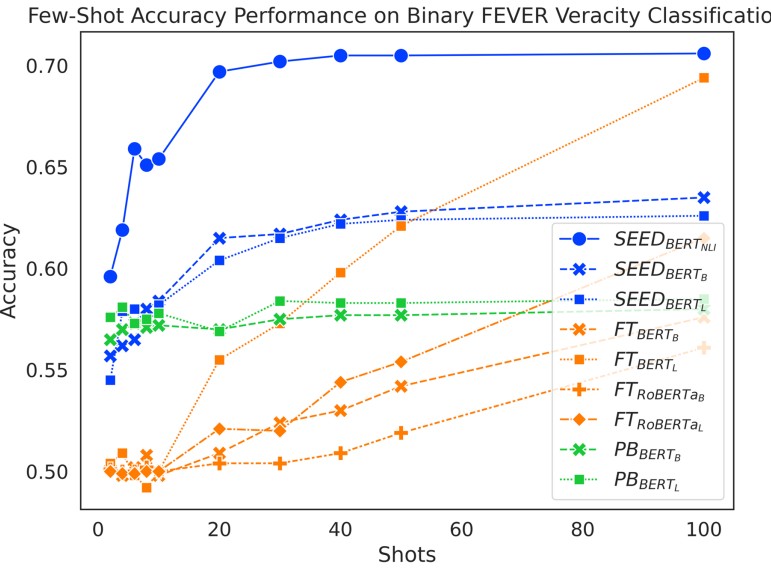

**Figure 2** **Comparison of few-shot accuracy performance on the binary FEVER dataset.**

## FEVER binary classification

### Experiment setup

For binary classification, we use the FEVER data provided by the original authors of the PB method (*Lee et al., 2021*) for fair comparison. The data contains 3,333 *"Support"* instances and 3,333 *"Not_Support"* instances[4]. For *n*-shot settings, we sample *n* instances per class as the train set, and use $3333 - n$ instances per class as the test set. We present experiments with all three methods (SEED, PB, FT).

### Results

As shown in Fig. 2, SEED achieves the overall best performance in few-shot settings. It suggests positive answers to our research questions: sentence embeddings from pretrained language models can be effectively utilised to compute semantic differences between claim-evidence pairs (RQ1) and they do contribute positively to the task of claim verification in few-shot settings (RQ2). When given fewer than 10 shots, the accuracy of the FT method remains low at around 50%, which is close to a random guess for a balanced, binary classification task. Meanwhile, $PB_{BERT_B}$, $PB_{BERT_L}$, $SEED_{BERT_B}$ and $SEED_{BERT_L}$ achieve similar results at around 57%. In 10-shot, 20-shot and 30-shot settings, SEED outperforms PB, which in turn outperforms FT. In 40-shot and 50-shot settings, $FT_{BERT_L}$ surpasses PB, although $FT_{BERT_B}$, $FT_{RoBERTa_B}$ and $FT_{RoBERTa_L}$ perform remarkably lower. In the 100-shot setting, $FT_{BERT_L}$ manages to outperform $SEED_{BERT_B}$ and $SEED_{BERT_L}$ and achieves similar performance as $SEED_{BERT_{NLI}}$. $FT_{BERT_B}$, $FT_{RoBERTa_B}$ and $FT_{RoBERTa_L}$ in the 100-shot setting failed to outperform SEED, despite that $FT_{RoBERTa_L}$ successfully outperformed PB. Overall, SEED with vanilla pre-trained language models outperforms both baselines from 10-shot to 50-shot settings. In addition, $SEED_{BERT_{NLI}}$ always achieves the best performance up to 100 shots.

[4] The *"Not Support"* is obtained by sampling and merging original instances from both *"Contradict"* and *"Neutral"*.

Interestingly, the increase of shots has very different effects on each method. SEED experiences significant accuracy improvement as shots increase when given fewer than 20 shots; the performance boost slows down drastically afterwards. Starting with reasonably high accuracy, PB achieves a mild performance improvement when given more training samples. When given fewer than 10 shots, the FT method doesn't experience reliable performance increase over training data increase; it only starts to experience linear performance boost after 10-shots.

## FEVER three-way classification

### Experiment setup

We use 3,333 randomly sampled instances for each class out of *"Support"*, *"Contradict"* and *"Neutral"* from the original FEVER test set as the total dataset for our experiment. For n-shot setting, we sample $n$ instances per class as the train set, and use $3333 - n$ instances per class as the test set. In these experiments we compare SEED and FT, excluding PB as it is only designed for binary classification.

### Results

Figure 3 shows a general trend to increase performance as the amount of training data increases for both methods. When given 10 or fewer shots, SEED shows significant performance advantages. When given between two and 10 shots, performance of fine-tuned models fluctuates around 33%, which equals to a random guess. Meanwhile, SEED achieves significant accuracy improvement from less than 40% to around 55% with vanilla pre-trained language models. In this scenario, the performance gap between the two methods that use the same model base ranges from 6% to 26%. With 20 shots, SEED with vanilla pre-trained language models significantly outperform $FT_{BERT_B}$, $FT_{RoBERTa_B}$ and $FT_{RoBERTa_L}$, although $FT_{BERT_L}$ managed to achieve similar results. With 30 shots, SEED with vanilla pre-trained language models reaches its performance peak at around 60% and $SEED_{BERT_{NLI}}$ peaks at around 68%. Given 30 or more shots, SEED slowly gets surpassed by the FT method. Specifically, $FT_{BERT_L}$ surpasses SEED with vanilla pre-trained language models using 30 shots, while $FT_{RoBERTa_L}$ and $FT_{BERT_B}$ only achieve a similar effect with 100 shots. However, $FT_{RoBERTa_B}$ never outperforms SEED within 100 shots. In addition, $SEED_{BERT_{NLI}}$ has substantial performance advantages when given fewer than 10 shots, despite being outperformed by $FT_{BERT_L}$ at 40 shots. Overall, SEED experiences a performance boost with very few shots, whereas the FT method is more demanding, whose performance starts to increase only after 10 shots. Like performance on binary FEVER, performance on three-way FEVER also suggests positive answers to our research questions: semantic differences between claim-evidence pairs can be captured by utilising sentence embeddings (RQ1) and positive contributions to the task of claim verification in few-shot settings are observed (RQ2).

Interestingly, $SEED_{BERT_B}$ outperforms $SEED_{BERT_L}$ starting from six shots. This performance difference within SEED further results in another interesting observation: $SEED_{BERT_B}$ achieves better overall accuracy than $FT_{BERT_L}$ at 10 shots.

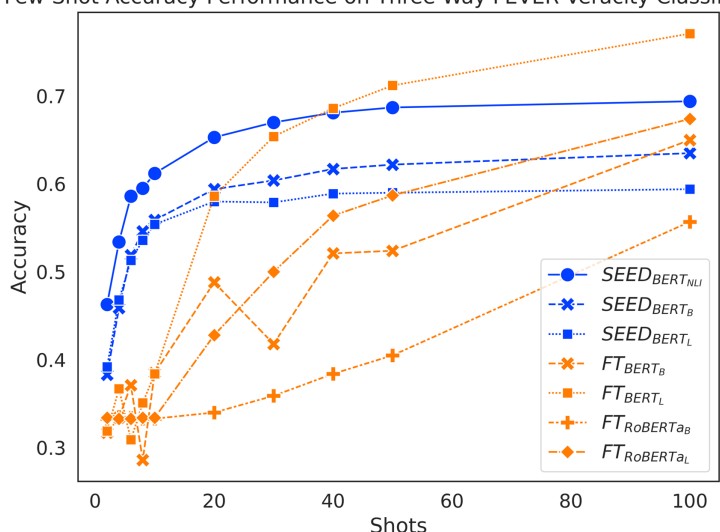

Few-Shot Accuracy Performance on Three-Way FEVER Veracity Classification

**Figure 3 Comparison of few-shot accuracy performance on the FEVER dataset.**

## SCIFACT three-way classification

### Experiment setup

The SCIFACT dataset is much smaller than the FEVER dataset, originally with only 809 claims for training and 300 claims for development (the test set being withheld for a shared task is not yet available at the time of writing). For each n-shot setting, we randomly sample *n* instances for each class out of *"Support"*, *"Contradict"* and *"Neutral"*, which are used as the train set. Given the imbalanced nature of the development set (*i.e.*, 138, 114 and 71 pairs for each class), we randomly sample 70 instances for each class in the development set and use them for evaluation. We again compare SEED and FT in these experiments.

### Results

Figure 4 shows again an expected increase in performance for both methods as they use more training data. Despite taking a bit longer to pick up, SEED still starts its performance boost early on. Increasing from 2 to 10 shots, SEED gains a substantial increase in performance. In addition, the FT method performs similarly to a random guess at around 33% accuracy when given 10 or fewer shots. When given 20 shots, FT still falls behind SEED, which differs from the trend seen with the FEVER three-way claim verification. $SEED_{BERT_B}$ and $SEED_{BERT_L}$ peak at around 45%, while $SEED_{BERT_{NLI}}$ peaks at around 50% with only 20 shots. At 30-shots and 40-shots, SEED still shows competitive performance, where $FT_{BERT_L}$ outperforms two of the SEED variants, but still falls behind $SEED_{BERT_{NLI}}$. $FT_{RoBERTa_L}$ outperforms SEED with vanilla BERT models at 50-shots and $FT_{BERT_B}$ and $FT_{RoBERTa_B}$ achieves that at 100-shots. Similarly, performance on SCIFACT dataset leads to positive answers to our research questions: sentence embeddings from pretrained language models can be effectively utilised to compute semantic differences and make positive contributions to few-shot claim verification task.

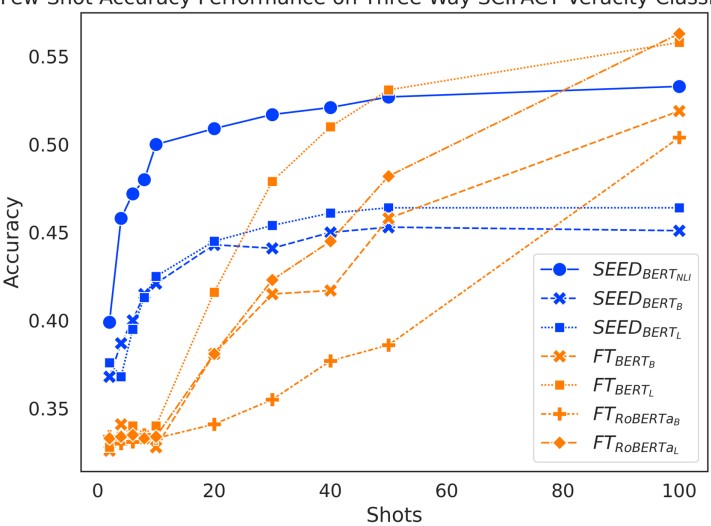

Few-Shot Accuracy Performance on Three-Way SCIFACT Veracity Classification

**Figure 4 Comparison of few-shot accuracy performance on the SCIFACT dataset.**

The accuracy scores on the SCIFACT dataset are noticeably lower than on the FEVER dataset. The FT method is again more demanding on the number of shots and experiences a noticeable delay to overtake SEED, more so on SCIFACT than on FEVER. This highlights the challenging nature of the SCIFACT dataset, where SEED still remains the best in few-shot settings.

## Classwise F1 performances

We present classwise F1 performance here for further understanding of the results. Figure 5 sheds light on addressing the task of FEVER binary claim verification. Both SEED and FT method gain improved performance on both classes with more data. The SEED method and PB method have significant performance advantages on the "Support" class, when given 10 or fewer shots. Despite that the PB method initially achieves very high performance on the "Support" class at around 60%, it then experiences a performance drop and ends at around 55% for BERT-base and 58% for BERT-large.

Figures 6 and 7 show consistent classwise performance patterns in tackling three-way claim verification on both FEVER and SCIFACT. Both figures indicate that SEED has better overall performance in all three classes when given fewer than 20 shots, where performance on the "Support" class always has absolute advantages over the FT method and performance on the "Neutral" class experiences the biggest boost. At around 20-shots the FT method starts to overtake largely due to improved performance on the "Neutral" class. Interestingly, within SEED, $SEED_{BERT_B}$ outperforms $SEED_{BERT_L}$, which in turn outperforms $SEED_{BERT_{NLI}}$.

Furthermore, classwise F1 performance also sheds light on the interesting SEED performance difference noted previously: $SEED_{BERT_B}$ outperforms $SEED_{BERT_L}$ in three-way claim verification with noticeable margin on FEVER three-way claim verification. Figure 6 shows that $SEED_{BERT_B}$ has clear performance advantages over $SEED_{BERT_L}$ on the

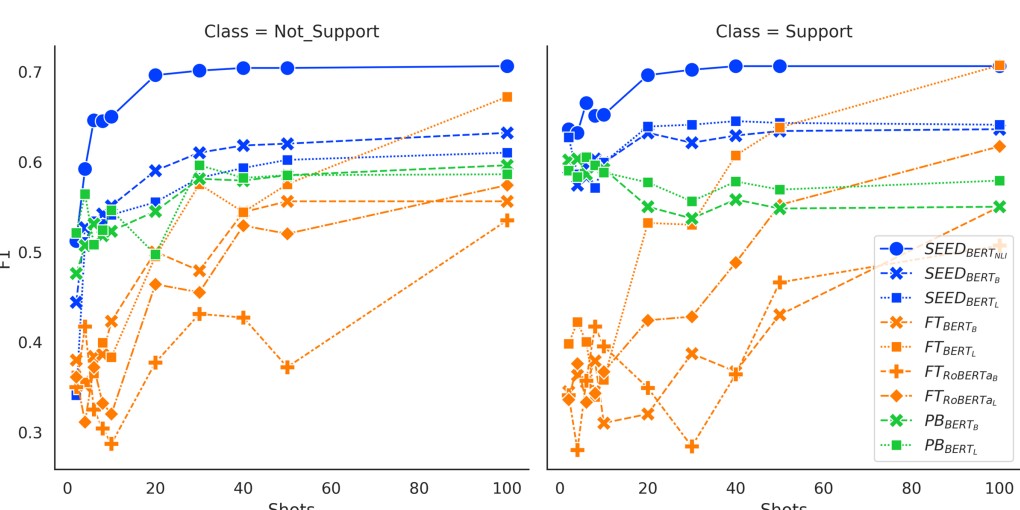

**Figure 5** Comparison of few-shot classwise F1 performance on the binary FEVER dataset.

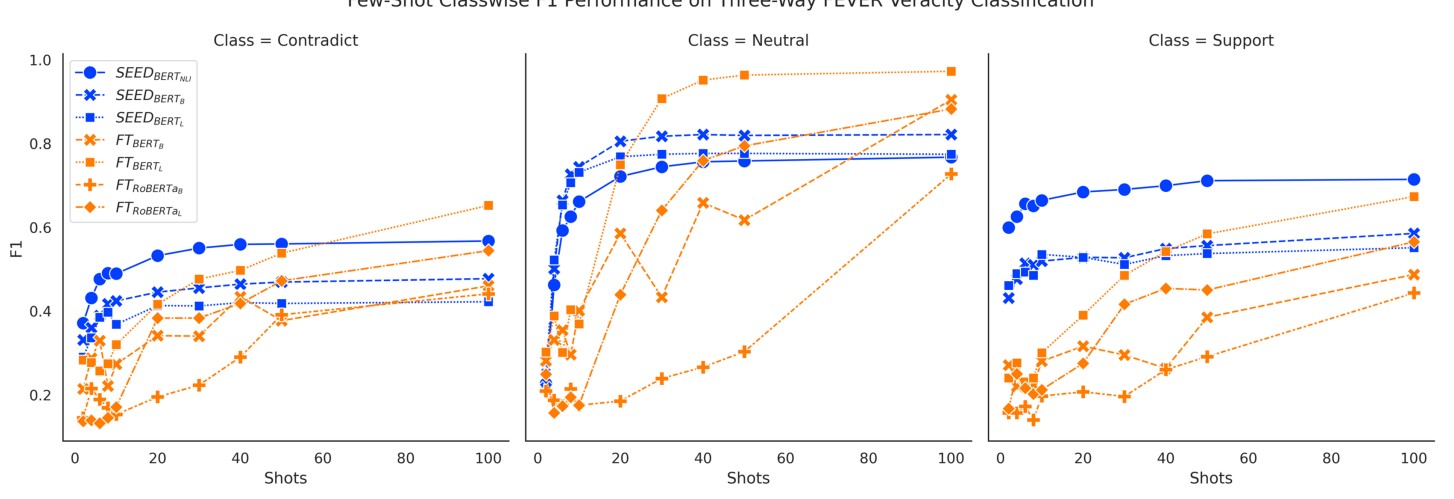

**Figure 6** Comparison of few-shot classwise F1 performance on the FEVER dataset.

"Contradict" and "Neutral" classes on FEVER three-way claim verification, which may be the main cause of the performance difference. When conducting binary claim verification on FEVER where the "Contradict" and "Neutral" classes are merged together, the performance advantages from $SEED_{BERT_B}$ over $SEED_{BERT_L}$ are trivial. Otherwise, $SEED_{BERT_B}$ does not outperform $SEED_{BERT_L}$ on the SCIFACT dataset as shown in Fig. 4. Meanwhile, Fig. 7 does not demonstrate $SEED_{BERT_B}$'s performance advantages on distinguishing the "Contradict" and "Neutral" classes on SCIFACT. We conjecture that $SEED_{BERT_B}$ is better at capturing simple differences between "Contradict" and "Neutral" classes while $SEED_{BERT_L}$ is better at capturing complex differences due to their size

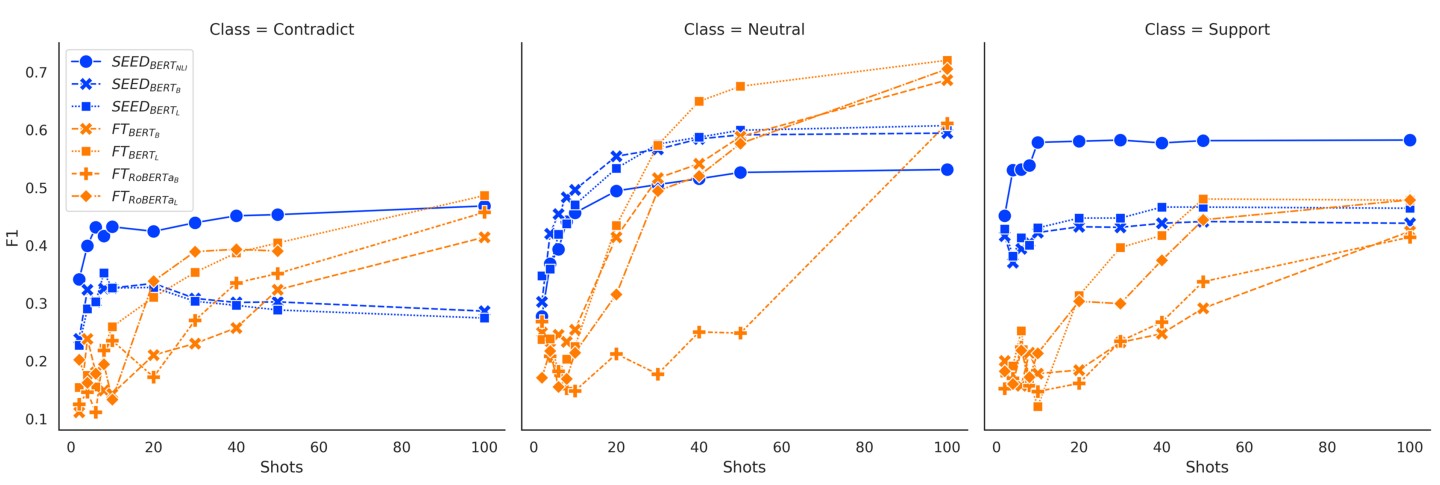

**Figure 7** Comparison of few-shot classwise F1 performance on the SCIFACT dataset.

**Table 2 Statistical significance test results in 20-shot setting.**

|  | Binary FEVER | FEVER | SCIFACT |
|---|---|---|---|
| $p$ value | $4e^{-38}$ | $1e^{-110}$ | 0.00679 |

difference. Given that FEVER is a synthetically generated dataset, it is to be expected that it includes more cases of simpler differences.

In general, classwise F1 performance shows consistent performance patterns with overall accuracy performance. The SEED method has significant performance advantages when given 10 or fewer shots in all classes. The PB method has very good performance on predicting the "Support" class initially but struggles to improve with more data. The FT method has underwhelming performance on all classes when given very few shots and gain big improvements over training data increase, especially on the "Neutral" class.

## Statistical significance

We present statistical significance test results conducted based on McNemar's test to demonstrate robustness of SEED, compared with FT. For demonstration purposes, results are calculated in 20-shot setting with the sampling seed set as 123 across three task formulations. For fair comparison, we use vanilla BERT-base as the base model for both SEED and FT methods.

Table 2 presents $p$ values. The $p$ values are always smaller than 0.005, indicating statistical significance for performance improvements obtained by SEED across three task formulations. Noticeably the $p$ value calculated on binary FEVER and three-way FEVER are much smaller than the $p$ value on SCIFACT, which suggests that the performance advantages are less significant. It correlates well with task difficulty: SCIFACT is more challenging than FEVER. Overall, SEED achieves significant improvements over FT in 20-shot setting.

# POST-HOC ANALYSIS

## Impact of shot sampling on performance

Random selection of $n$ shots for few-shot experiments can lead to a large variance in the results, which we mitigate by presenting averaged results for 10 samplings. To further investigate the variability of the three methods under study, we look into the standard deviations.

Figure 8 presents the standard deviation distribution on Binary FEVER claim verification, which is largely representative of the standard deviations of the models across the different settings. We only analyse configurations that utilise BERT-base and BERT-large for direct comparison across methods. Overall, PB always has the lowest standard deviation, which demonstrates its low performance variability across random sampling seeds. Combined with the initial performance boost of SEED in Fig. 2, the high standard deviation in the beginning implies that the SEED method is able to learn from the extremely limited number of training data and therefore experiences performance fluctuations due to different few-shot samples. Meanwhile, when given 10 or fewer shots, FT's accuracy performance remains close to random guess (see Fig. 2) and its standard deviation remains low (see Fig. 8). The low performance and the insensitivity to different sampling seeds indicates in this scenario that the FT method is not able to effectively learn from the extremely limited number of data. As the number of training samples further increases beyond 10 shots, the standard deviation of SEED drastically decreases and its performance experiences a boost until it converges at around 40 shots. After the initial performance boost, the SEED method shows robustness to random sampling. When given more than 10 shots, the standard deviations of FT surpass SEED with a large margin and its accuracy performance starts to experience a boost, which indicates that the FT models are able to learn from the given samples in this scenario. However, the FT models do not converge within the first 100 shots, which leads to high standard deviation within the range from 20-shots to 100-shots and they remain vulnerable to random sampling in few-shot settings.

In short, PB is the most robust method to sample variations, despite underperforming SEED on average; SEED is still generally more robust to random sampling and has higher learning capacities than the FT method in few-shot settings.

## Why does SEED plateau?

As presented in the Results section, the performance improvement of SEED becomes marginal when given more than 40 shots. Given that SEED learns mean representative vectors based on training instances for each class, the method likely reaches a stable average vector after seeing a number of shots. To investigate the converging process of representative vectors, we measure the variation caused in the mean vectors by each additional shot added. Specifically, for values of $n$ ranging from 2 to 100, we calculate the Euclidean distance between n-shot relation vectors and $(n - 1)$-shot representative vectors, which measures the extent to which representative vectors were altered since the addition of the last shot. Figure 9 depicts the converging process with FEVER three-way claim

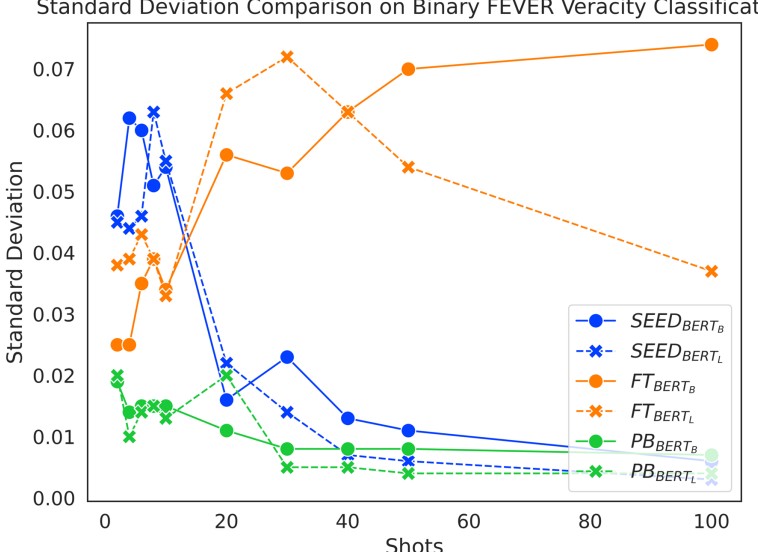

**Figure 8 Standard deviation comparison on binary FEVER claim verification.**

verification. Across three different model bases, the amount of variation drops consistently for larger numbers of n shots, with a more prominent drop for $n = \{2-21\}$ and a more modest drop subsequently. From a positive angle, this indicates the ability of SEED to converge quickly with low demand on data quantity. It validates the use of semantic differences for verification and highlights its efficiency of data usage in few-shot settings. From a negative angle, it also means that the method stops learning as much for larger numbers of shots as it becomes stable, *i.e.*, it is particularly useful in few-shot settings.

The curves of BERT-base and BERT-large largely overlap with each other, while the curve of BERT-nli does not conjoin until convergence. It corresponds well with the overall performance advantages of utilising BERT-nli as presented in the Results section. It implies that using language models fine-tuned on relevant tasks allow larger impact to be made with initial few shots. Future work may deepen the explorations in this direction. For example, using a model fine-tuned on FEVER claim verification to address SCIFACT claim verification.

## DISCUSSION

With experiments on two- and three-class settings on two datasets, FEVER and SCIFACT, SEED shows state-of-the-art performance in few-shot settings. With only 10 shots, SEED with vanilla BERT models achieves approximately 58% accuracy on binary claim verification, 8% above FT and 1% above PB. Furthermore, SEED achieves around 56% accuracy on three-way FEVER, while FT models underperform with a 38% accuracy, an absolute performance gap of 18%. Despite the difficulty of performing claim verification on scientific texts in the SCIFACT dataset, SEED still achieves accuracy above 42%, which is 9% higher than FT. When utilising BERT-nli, SEED consistently achieves improvements with 10 shots only: 15% higher than FT and 8% higher than PB on FEVER

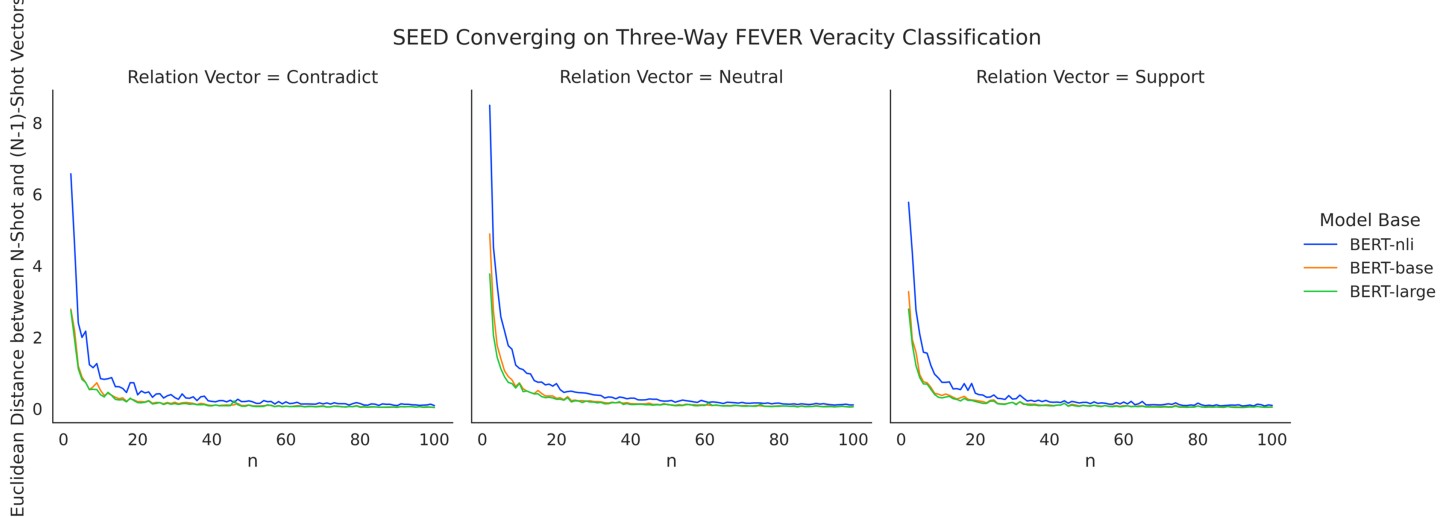

**Figure 9 SEED converging on three-way FEVER claim verification with increasing number of shots.**

binary claim verification; 23% higher than FT on FEVER three-way claim verification and 17% higher than FT on SCIFACT three-way claim verification. Further, detailed analysis on classwise F1 performance also shows that improved performance is consistent across classes.

Our experiments successfully address our research questions that sentence embeddings from pre-trained language models can be effectively utilised to compute pairwise semantic differences between claims and their associated evidences with limited labelled instances (RQ1). The proposed method leads to positive contributions with improved performance on the task of claim verification in few-shot settings (RQ2). In comparison with PB, SEED has better learning capacities, higher few-shot performance, and most importantly, it is more flexible for doing multi-way claim verification, enabling in this case both two-class and three-class experiments. With respect to FT, SEED is better suited and faster to deploy in few-shot settings. It is more effective regarding few-shot data usage, generally more robust to random sampling, and it has lower demand on data quantity and computing resources.

The main application scenario of SEED is few-shot pairwise classification, *i.e.*, when the input involves text pairs. While we have demonstrated its effectiveness on few-shot claim verification, future work may study the effectiveness of SEED on other pairwise classification tasks, *e.g.*, natural language inference, stance detection, knowledge graph completion and semantic relation classification between documents. Furthermore, SEED also offers the potential to be used for annotation quality evaluation: SEED is sensitive to data sampling within 10 shots and it may be utilised as a good metric to determine whether the annotated data is of high quality or not with only a few samples. Moreover, SEED can be applied to do task difficulty estimation: SEED's few-shot performance on SCIFACT is significantly lower than FEVER, which correlates well with the fact the

SCIFACT is more challenging than FEVER. In future studies, one may conduct few-shot experiments without gradient update using SEED on a new dataset and a familiar dataset to gain valuable initial understanding on the difficulty of the new dataset.

While SEED demonstrates the ability to learn representative vectors that lead to effective claim verification with limited labelled data and computational resources, its design remains simple and its performance plateaus with larger numbers of shots. Future studies may further develop the method by utilising more advanced sentence embeddings. For example, while our proposed SEED calculates mean values of all tokens for sentence embeddings, future work may obtain syntactically aware sentence embeddings by calculating weighted average values with reference to syntactic parse trees. In addition, further exploration into SEED's potential to further improve its performance when more training samples are observed would also be a valuable avenue of future research. One possibility to achieve this could be by extending SEED with the use of gradient descent.

## CONCLUSIONS

We have presented an efficient and effective SEED method which achieves significant improvements over the baseline systems in few-shot claim verifications. By comparing it with a perplexity-based few-shot claim verification method as well as a range of fine-tuned language models, SEED achieves state-of-the-art performance in the task on two datasets and three different settings. Given its low demand on labelled data and computational resources, SEED can be easily applied, for example, to new domains with limited labelled examples. Future research may further extend SEED with more sophisticated sentence embeddings. While our focus here has been on few-shot learning, future research could focus on building a capacity to more effective learning from larger numbers of training samples.

## ACKNOWLEDGEMENTS

This research utilised Queen Mary's Apocrita HPC facility. http://doi.org/10.5281/zenodo.438045.

### Funding

This work was supported by the Engineering and Physical Sciences Research Council (Grant EP/V048597/1). Xia Zeng is funded by the China Scholarship Council (CSC). The funders had no role in study design, data collection and analysis, decision to publish, or preparation of the manuscript.

### Grant Disclosures

The following grant information was disclosed by the authors:
Engineering and Physical Sciences Research Council: EP/V048597/1.
China Scholarship Council.

## Competing Interests

Arkaitz Zubiaga is a Section Editor for PeerJ Computer Science.

## Author Contributions

- Xia Zeng conceived and designed the experiments, performed the experiments, analyzed the data, performed the computation work, prepared figures and/or tables, authored or reviewed drafts of the article, and approved the final draft.
- Arkaitz Zubiaga conceived and designed the experiments, authored or reviewed drafts of the article, and approved the final draft.

## Data Availability

Both the data and code are available at GitHub: https://github.com/XiaZeng0223/seed; Zeng, Xia, & Zubiaga, Arkaitz. (2022). Aggregating pairwise semantic differences for few-shot claim verification. https://doi.org/10.5281/zenodo.7064913.

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
