# Peer review of "Aggregating pairwise semantic differences for few-shot claim verification"

_PeerJ Computer Science, doi:10.7717/peerj-cs.1137_

## Round 0.1 · original submission · Minor Revisions

The reviewers agree your article offers an interesting and valuable contribution. We look forward to receiving a revised version addressing the reviewers' comments.

Reviewer 1 ·

Basic reporting

** Points of Strength
- The tackled problem is important and interesting.
- The paper is well-written and well-organized.
- Figures are clear and relevant.
- The article is self-contained.

** Major Issues:
- While there are many references, there is not enough depth in citing/positioning many of them (check first paragraph in BACKGROUND section).
- I'd add at least one real example to motivate the problem, and another to motivate/show the proposed method.

** Minor Issues:
- There are few typos (line 9: consists in --> consists of, 23: consists in --> consists of, 265: direction --> direct, 295: allow --> allows), and few missing section references (lines 278, 294) though.
- I only have one issue with Fig. 1, as colors are not indicative (or at least explained).

Experimental design

** Points of Strength
- The paper presents original research within the scope of the journal.
- Conducted experiments on two different datasets.
- Compared performance with SOTA baselines.
- Used 10 random samples and 4 runs per experiment.
- Comprehensive experiments.
- Proposed method is simple and clearly described.

** Major Issues
- Research questions are not stated at all.
- No statistical significance tests were performed.

Validity of the findings

** Points of Strength
- Datasets used are publicly available.
- Conclusions are well stated and supported by the figures.

** Major Issues:
- Analysis of the results is shallow with no depth in some positions. It's mainly describing the results, not analyzing them. For example, there is no justification on why BERT-Base outperformed BERT-Large, why SD of FT is getting larger with more shots?
- The discussion section is mainly a repeated summary of the results. It should discuss impact of the results, failure analysis, etc.

** Minor Issues:
- The conclusion section gives very little direction on potential future work.

Reviewer 2 ·

Basic reporting

The paper is well written and easy to read.
It provides sufficient field background and context on fact verification and corresponding few-shot setting.
The article structure, figures, and data can support the paper's hypotheses.

Experimental design

The experiments for few-shot effectiveness on FEVER and SCIFACT are designed well and the baselines are selected and compared properly.
The method is simple but effective for fact-checking tasks under limited labeled resources, so the research question is meaningful.
The major part of the method is described sufficiently.
one thing not very clear to me is how the sentence representations are obtained from e.g. BERT model. Is it taking the 768 dimension vectors from the CLS token as sentence representation or averaging pooling the token representations from the last layer output? In the paper, the author stated they use the sentence-transformers toolkit, but it would be better to clearly describe how the sentence representation generated from BERT/ Roberta

Validity of the findings

The findings have good validity given that the authors agreed to release the code to reproduce the results.

---

## Round 0.2 · Minor Revisions

While you responded to all of the reviewers' comments, reviewer 1 is not satisfied with some of the responses, as shown in the comments. We encourage you to revise the manuscript addressing these comments and resubmit it. If any of these suggestions cannot be included in the revised manuscript, provide a justification in your rebuttal letter.

Reviewer 1 ·

Basic reporting

- Thanks for fixing the typos.
- About the background section: Authors didn't change the section. The high-level overview is just listing a large bunch of references without proper context. They should have been better positioned in the respective paragraphs with enough context to show how they are related, or removed to give space to the two main sub-problems to be reviewed in a better depth.
- About adding real examples: Authors didn't add any new examples. It is not clear if the already-given example is real or not, without any citation or source. Also, a small sample of examples to motivate the proposed solution can be selected/drawn from the datasets used in the experiments.
- About Fig.1: Authors made no change to the figure (only changed the description). It is not clear to me how the colors became indicate by mapping them to the vectors in the description; they are still not indicative/meaningful. For example, the representative vectors still have two different colors in the figure, for no obvious reason.

Experimental design

- About RQs: Thanks for adding the two RQs; however, in the Discussion Section, you just directly answered/repeated the RQs without linking them to the specific experiments that show the answer. There should be earlier links to the RQs within the analysis of the results.

Validity of the findings

- About significance tests: The SD values don't tell us in your work if the observed differences were observed by chance or not (i.e., if the differences/improvements are meaningful or not). There are no confidence scores or intervals mentioned in the paper. No proper citations or even working links were given to justify the choice of not applying significance tests. A means that can tell us that *differences in performance* are statistically meaningful is expected.

Reviewer 2 ·

Basic reporting

no further comments. The paper is clearly written and easy to follow. The issues mentioned by first-round reviewers are properly solved.

Experimental design

no further comments. The issues mentioned by first-round reviewers are properly solved.

Validity of the findings

the method is simple and effective for the few shot setting also has novelty to the task. The results of BERT-large get lower results than BERT-base are interesting, and the analyses suggested by the other reviewer are very meaningful and I think it gets solved in a good shape. The conclusions are well stated and the results are reproducible given that the author ready to share the code.

---

## Round 0.3 · accepted · Accept

Thank you for your contribution to PeerJ Computer Science and for systematically addressing all the reviewers' suggestions. All the reviewers are now happy with the current version and recommend its publication. Congratulations!

Reviewer 1 ·

Basic reporting

Revised version is somewhat satisfying.

Experimental design

Revised version is somewhat satisfying.

Validity of the findings

Revised version is somewhat satisfying.